# Nudging Consumers toward Healthier Food Choices: A Field Study on the Effect of Social Norms

**Diogo Gonçalves [1,\*], Pedro Coelho [2], Luis F. Martinez [2]** and **Paulo Monteiro [3]**

1 DEG-IST—Departamento de Engenharia e Gestão, Instituto Superior Técnico, Universidade de Lisboa, Av. Rovisco Pais, 1049-001 Lisboa, Portugal

2 Nova School of Business and Economics, Campus de Carcavelos, Universidade Nova de Lisboa, Rua da Holanda, 1, 2775-405 Carcavelos, Portugal; pedro42coelho@gmail.com (P.C.); luis.martinez@novasbe.pt (L.F.M.)

3 ESTeSL.IPL—Lisbon School of Health Technology, Av. D. João II, Lote 4.69.01, 1990-096 Lisboa, Portugal; paulo.monteiro@estesl.ipl.pt

\* Correspondence: diogoagoncalves@tecnico.ulisboa.pt

**Abstract:** Food choices influence the health of individuals, and supermarkets are the place where part of the world population makes their food choices on a daily basis. Different methods to influence food purchasing habits are used, from promotions to food location. However, very few supermarket chains use social norms, the human need to conform to the perceived behavior of the group, to increase healthy food purchase habits. This research seeks to understand how a social norm nudge, a message conveying fruit and vegetable purchasing norms positioned in strategic places, can effectively change food choices. Using data from an intervention in a Portuguese supermarket, the fruit and vegetable purchase quantities of 1636 customers were measured over three months and compared with the corresponding period of the previous year. The results show that the nudge intervention positively affected those whose purchasing habits are categorized as less healthy, while those with healthy habits were slightly negatively affected. Moreover, a follow-up inferential statistical analysis allows us to conclude that applying this intervention at a larger scale would deliver significant financial results for the supermarket chain in which the study took place, by decreasing the costs related to produce perishability while simultaneously improving the health of the consumer and the sustainability of the planet.

**Keywords:** nudging; behavioral science; health; food choices; sustainability; social norms



## 1. Introduction

Due to the fast-paced modernization and technological advances of our society, we currently live in a world where computers, cars, and devices can facilitate our lives but, on the other hand, this also results in a lack of mobility and physical activity [1].

The World Health Organization (WHO) states that the lack of physical activity coupled with practices of unhealthy diet have led to a burgeoning of health diseases such as diabetes, heart diseases, strokes, various types of cancers, overweight, and obesity. Society is consuming foods high in energy, fats, sugar, and salt, while not eating enough "green" foods such as fruits and vegetables [2].

This is most evident in the Portuguese context. According to the National Food, Nutrition, and Physical Activity Survey of the Portuguese general population, conducted from October 2015 to September 2016 [3], the overall consumption of fruits and vegetables is quite below the proportion of the recommended daily food intake for these food groups, according to the Portuguese Food Wheel Guide: −9% for vegetables (14% vs. 23%) and −7% for fruits (13% vs. 20%) while the "Meat, fish, and eggs" food group consumption is 12% higher relative to the recommendations (17% vs. 5%) [4]. When comparing the consumption profile of fruits and vegetables, by the Portuguese population, to the recommended intake

by the WHO (at least 400 g/day of these foods, equivalent to five or more portions per day), the prevalence of individuals who do not adhere is 56% [4].

According to the "*First Portuguese Health Examination Survey*" in 2015, the Portuguese adult population was determined to have overweight and obesity levels of 39.1% and 28.6%, respectively, putting it among the highest prevalence in Europe, comparable with England and Scotland. Researchers in this study advise that public health interventions are needed, "namely throughout health literacy strategies for promoting awareness of the health hazards" [5].

These interventions have been promoted by public health organizations, governments, and consumers through either legal regulations and economics measures or through education and industry codes [6]. Despite educational efforts and investments to increase communication regarding this important topic, most of the interventions were only able to create awareness, rather than change individuals' behaviors toward a healthier lifestyle [7]. In recent years, these interventions have been designed based on behavioral economic principles that change environments and alternatives because individuals acknowledge the need to change their diet and lifestyles but, due to the "vast availability of unhealthy foods, health illiteracy or self-regulatory skills", they are not able to fulfill this goal [8]. Supermarkets and retail stores account for most food purchases and represent a context that can heavily influence purchase and eating habits [9]. When in a supermarket or grocery store, consumers are faced with several options to choose from, calling for decisions to be fast and unconscious [10].

Decision-making processes are not totally rational and are heavily influenced by psychological biases (e.g., present-based biases, choice presentation formats) and fast and unconscious impulses [10]. Recent studies have shown that methods targeted at consumers' minds, instead of those that limit consumers' choices, seem to have a greater impact on improving the effectiveness of health campaigns [11]. These methods are often associated with the term "*nudge*", which refers to changing people's behavior without the constraint of options [12]. Because the environment in which individuals make choices can be altered and influence the way the decision-making processes occur, nudging focuses on enabling and changing behaviors and decisions that are beneficial for society (e.g., public health) rather than delivering information or changing the society's values system [12].

It is important to understand how nudging can affect behaviors and what positive outcomes it could deliver for society in the context of health. Additionally, supermarkets are the place where most of the consumers in modern societies make their regular food purchase choices. Thus, the way they are structured and how it influences food choices is a very important research topic. Prior studies on the impact of social norms on healthy food behavior have mainly addressed an academic food court context [13]; thus, our research aim seeks to extend this research to a real supermarket environment (i.e., from a major international food retailer). Accordingly, this study examines how nudges can impact different customer segments' purchasing behavior regarding healthy food. The impact of nudges in a specific food retail customer segment (i.e., women's food choices) has already been tested before [14] but, to our best knowledge, the type of segmentation used in our research (i.e., according to past purchases) is innovative. The intervention consisted of introducing a simple nudge message in the context of a supermarket. Moreover, it also investigates how effective this nudging strategy can be in comparison to price reductions and speculates if this type of intervention could be applied to all hypermarkets and supermarkets of the particular retail chain, and what the accumulated financial benefits might be.

## 2. Literature Review

### 2.1. Overview of Nudge

Nudges are defined as any aspect of the decision environment "( . . . ) that alters people's behavior in a predictable way without forbidding any options or significantly changing their economic incentives" [12] (p. 6). It is the adaptation of how information is

shown that enables nudges to bring great cost-effectiveness benefits, because studies have discovered that providing information does not necessarily result in effective changes in people's behavior [15,16]. The lack of positive results of providing only information tends to happen because people can be overwhelmed by the amounts of information, have a lack of self-control, or be influenced by different context aspects [15]. Therefore, nudges are usually simple and small changes in the "choice architecture" that represents the outside forces that guide one's decisions [17], and that will redesign the environment, with the goal of increasing the likelihood that a certain option will be chosen.

The concept of nudge is often linked to "libertarian paternalism", which is a policy approach that preserves the freedom of choice, although it uses mechanisms to guide people's directions toward desired society behaviors [18]. An ongoing debate about morality of both nudges and libertarian paternalism claim that these methods violate the principle of freedom of choice, although their popularity among decision makers and governments is on the rise [19,20]. Nudging seems to work best on the uninformed and uneducated part of society, where there is less access to information and understanding, and can be perceived as an unfair method because those who are informed and able to identify a nudge can free-ride the efforts of the majority of those who cannot. These cases tend to occur when policy makers endeavor to achieve common goods such as climate change mitigation and sustainability [20,21]. Thus, transparency is an important aspect to take into consideration—in fact, some authors accuse nudge of being manipulative, as it can be a real risk of power abuse on the part of governments [22].

Despite the contradictions, some authors [12] believe that nudge was created to promote behaviors that are beneficial for both individuals and society using effective interventions. Policy makers embrace two different ways of nudge usage: promote positive behaviors and increase their long-term effects and usage (e.g., the consumption of healthy food or the reduction of tobacco or alcohol consumption) or counteract the negative impact of other actors (e.g., business, media) and reduce their effects in the long term [23]. Although nudge interventions seem to be effective in the short term, one of the biggest limitations is that studies do not evaluate their long-term effectiveness [24].

Finally, nudges can be classified into four different types based on how they influence and enable individuals' choices: (i) *simplification and framing of information*—simplification intends to streamline information, simplifying processing capabilities and decision-making processes, while framing relies on how information is expressed, activating certain values and attitudes; (ii) *changes to the physical environment*—especially in the retail context, careful product placement is acknowledged to have a considerable impact on purchasing choices; (iii) *changes to the default policy*—standard options can determine individuals' choices because people often take the path of least resistance, allowing them to be greatly influenced by default choices; and (iv) *use of social norms*—socialization can have a huge impact on human behaviors because it shapes one's behaviors to mirror those of peers [25].

### 2.2. Social Norms in Nudge Interventions

Nudge interventions are often designed based on the social norms of a given context. The reasoning behind this is that individuals use their perceptions of peer norms as a yardstick when comparing their behaviors to those of their social peers [26,27], and the majority of individuals overestimate the prevalence of undesirable behaviors [28,29].

Therefore, nudge interventions that use social norms designed to reduce alcohol consumption, drug use, eating behaviors, gambling, and/or recycling are seen as an alternative to traditional marketing campaigns (e.g., information campaigns, fear inducing messages) [30]. The objectives of these campaigns are not only to reduce undesirable behaviors but also to correct the targets' misperceptions of the social norm using a descriptive norm [31]. Although some interventions have resulted in favorably altered behaviors, there are cases in which the social-norm campaigns had a boomerang effect that led to the increase of those undesirable behaviors [32,33].

Usually, these types of interventions provide descriptive normative information that refers to what is commonly done or is normal in each situation [34]. By comparing their behavior with norms, individuals who perform above the norm are expected to reduce the undesired behavior, but for individuals who already rank below the norm, this information might have a reversal effect and lead them to increase the undesirable behavior [35]. According to the focus theory of normative conduct [31], there is another type of social norm—injunctive norms—that also exerts a powerful influence on behavior. Injunctive norms can be described as the perceptions about what is accepted or disapproved of within a culture [36]. In the case of individuals who already rank below the descriptive norm, the use of injunctive norms (i.e., referring to what is accepted or disapproved of by society) can prevent them from increasing the undesirable behavior [37].

An experimental study undertaken in a California community was designed to investigate what effects a descriptive norm vs. injunctive plus descriptive norm would have on household energy consumption. Results showed that the descriptive norm would lead consumers above the norm to reduce consumption while those below would raise their consumption levels. On the other hand, when applying both norms, there was an insignificant increase in those below the norm and a decrease of those above. The authors concluded that applying both norms would reduce the boomerang effects [38].

A social norm similar to the one in the case above can be adapted to food retailing in order to change individuals' behaviors to increase or decrease a certain type of product. In our specific intervention, the goal was to raise the quantity of fruits and vegetables purchased by displaying a message regarding a consumption social norm in a supermarket context.

*2.3. Nudge in Health and Food Retail*

Many interventions aiming to increase the consumption of healthier foods through nudges have been tested and implemented in recent years. For example, a school cafeteria in New England (North America) asked their students—before they ordered their meals—whether they would have fruit or juice with their lunch, and the intervention resulted in 70% of students consuming one of those in opposition to 40% in the control group [39]. An intervention in a buffet restaurant in Denmark changed the sequencing design of its service combining and separating fruits and vegetables. The change resulted in an increase of self-served fruits and vegetables while reducing the total calorie intake [40]. Despite the rise in interest for these kinds of interventions, there are still a limited number of articles regarding direct food choices. Most articles and interventions focus on store interventions that influence consumers' food purchases [41].

Regarding a retail context, most of the interventions are simple changes to product information, physical placement, or environment aspects. In a school cafeteria, adding a descriptive label to the most popular items such as Zucchini Cookies to Grandma's Zucchini Cookies, resulted in a 27% increase in sales [42]. Relocating fruits and vegetables closer to the cashier instead of sweets resulted in a sales increase of those products [43]. A study demonstrated that US consumers were more likely to choose a healthy item from a shelf when it was positioned at the left when compared to the unhealthy items at the right side, due to display being congruent with their natural mental representation [44].

Most supermarket retailers are motivated to sell fruits and vegetables despite not knowing how to do it effectively [45]. Interventions that focus on making these products more convenient, attractive, and normal to select in opposition to unhealthy foods are more likely to be successfully implemented [46].

Convenience in the store is boosted by demonstrating how a healthy food can be incorporated into a cooking routine that is familiar for the consumer or placing the product in strategic places that have higher visibility [47]. For example, the largest tofu manufacturer in the US created an in-store campaign that taught consumers how to prepare and cook tofu in 10 min, leading to an increase in sales and shopper confidence [48]. Additionally, placing fruits and vegetables at the front entrance, where every customer needs to pass, increased a supermarket's sales by 8% compared to those that did not [49]. Attractiveness refers

to how attractive a healthier food can be when compared to an unhealthy one by either changing appearance or price [50]; rebates, coupons, and bundling can be used to increase fruit and vegetables sales, although sometimes it can also result in the increased sales of unhealthy foods in low-income households [51]. Displaying fruits and vegetables with a decorated frame around instead of piling them onto a flat table resulted in an increase in both attractiveness and sales [52]. Moreover, signage improves the selection of food that people believe to be normal or popular to purchase, as many experiments have shown [46]. For example, signs indicating that Harlem shoppers preferred chickpeas to other beans resulted in a 14% increase of its sales [53]. Additionally, shopping carts stating the average purchase of at least five fruits and vegetables increased those sales by 10% [54]. Moreover, a visual norm called the "half-plate rule" was used to create balanced meals, as half of your plate should have fruits and vegetables and the other half can be filled with whatever you want, encouraging shoppers to buy greater quantities of fruits and vegetables [46]. In another study, similar to the "half-plate rule", the "half-cart approach" was used, which consisted of dividing shopping carts into two partitions, instructing that one half be filled with fruits and vegetables and the other half with everything else. This experiment resulted in an 18% increase of fruit and vegetable purchases [55].

Overall, this study's goal is to test the impact of a social norms' message in terms of increasing the purchase of fruits and vegetables in a real-world retail food sales context (i.e., the supermarket of a multinational retail chain). Thus, our first hypothesis regarding the results of the study is:

**Hypothesis 1 (H1).** *A descriptive norm intervention in the store will increase the purchase of fruits and vegetables (compared to a period without intervention).*

The message created conveyed the behavior of the healthiest families visiting the supermarket. Eleven fruits and vegetables were chosen because this number of items/products refers to the threshold that includes the top 10% receipts (i.e., 90th percentile) for the actual consumption of these products. A cognitive dissonance-inducing expression (i.e., "E você?"—"And you"?), used to nudge the other clients to conform to the behavior of the top 10% clients in terms of fruit and vegetable purchase profile, was also included. Accordingly, the study goals were not only to test the overall impact of the message, but also the segmented impact according to the previous purchase habits of the clients (i.e., low, mid, and high-purchase buyers), according to the month of testing (i.e., first, second, and third). This would allow us to understand if the impact of the nudge varied according to the shoppers' habits of fruit and vegetable consumption, and according to the familiarity of the intervention (e.g., would it lose impact as time goes by or become more familiar). Accordingly, our second hypothesis regarding the results of the study is:

**Hypothesis 2 (H2).** *Fruit and vegetable purchases will increase more among soft/medium buyers (compared to heavy buyers with high fruit/vegetable purchase).*

## 3. Materials and Methods

### 3.1. Participants and Procedures

This intervention took place in *Amoreiras Shopping Center* (located in Lisbon) at *Auchan*, a French supermarket chain operating in Portugal, and it was deployed by Nudge Portugal with the goal of increasing the purchase of fruits and vegetables. It ran for a period of three months (May, June, and July of 2019).

The procedure was as follows: Auchan customers were exposed to a social norm in the form of *signage* when taking a shopping cart, with the goal of changing their shopping habits to healthier ones. Understanding the way in which these kinds of nudges influence customers, and their effectiveness, can help companies to achieve higher margins on a specific type of product without the need to invest large amounts of money in marketing

campaigns. At the same time, in this specific case, the nudges made customers feel healthier by guiding their choices to what is perceived as a norm.

All the data were gathered from the customer-loyalty cards from that specific store. For the data to be as close as representative to reality, data were gathered only for customers who purchased at least one fruit or vegetable and used their customer card for payment at the checkout. For comparison purposes, the period chosen to be fully analyzed was from 25 April 2018 to 25 May 2018, when there was no intervention, and 25 April 2019 to 25 July 2019, when the intervention was in place. It is thereby possible to compare the exact same customers for the exact same period, enhancing the value of the data. Furthermore, the result of the intervention is also analyzed, including the changes over the three months examined.

After filtering, the sample was comprised of 1636 customers who purchased at least one fruit or vegetable during the study period. It was taken into consideration that one client could have multiple tickets, and those data were gathered as well. Apart from the number of tickets, number of fruits and vegetables, and variety of products, no more information pertaining to the shoppers was collected.

Clients who obtained shopping carts were immediately exposed to the message placed between the handles, "The healthiest families of this store purchase at least eleven fruits and vegetables per visit. And you?" (in Portuguese, "As famílias mais saudáveis desta loja, escolhem pelo menos 11 Frutas e Legumes por visita. E você?") (Figure A1), along with a picture of a cart with apples, pears, oranges, and vegetables inside. The same display was also strategically placed near the scales, as it was close to the moment of decision on how many fruits and vegetables to buy, as well as near the location to pick up baskets. As with most nudge interventions, this represents a very small cost to the store and an elevated awareness.

*3.2. Segmentation and Variables*

In order to analyze and make conclusions from the purchase behaviors, it is crucial to segment the data into different purchase groups, due to the considerable differences between the amounts of articles purchased. As seen in other studies, performing an overall analysis of the intervention will not provide accurate and significant insights to the study because different groups will be differently impacted by the nudge exposure. It was found that splitting the analysis into three different categories could provide interesting insights from the study and enhance the quality of its recommendations.

Therefore, the 1636 customers were segmented based on the total number of fruits and vegetables purchased; that is, the sum of all fruits and vegetables on each of the receipts in that period of time. The first group, called *SoftBuyers,* included the clients who purchased fewer than four articles. The second group, *MediumBuyers*, included clients who purchased between four and ten fruits and vegetables. The largest group, *HardBuyers*, included those who purchased more than ten articles. With this segmentation, it is possible to analyze each type of consumer, allowing us to identify variations in purchase behavior as well as in the type of product purchased (see Table 1).

**Table 1.** Sample segmentation according to previous purchase frequency.

|  | May | June | July | Total (3 Months) |
|---|---|---|---|---|
| Soft (<4) | 670 (40.9%) | 695 (40.6%) | 614 (39.9%) | 1157 (39.1%) |
| Medium (4 to 10) | 420 (25.7%) | 447 (26.1%) | 405 (26.3%) | 830 (28.1%) |
| Hard (>10) | 546 (33.4%) | 569 (33.3%) | 521 (33.8%) | 969 (32.8%) |
| Total | 1636 | 1711 | 1540 | 2956 |

The study focused on the analysis of five variables, three of which were "raw" variables and two that were derived from the data gathered. The variables *Receipts19* and

*Receipts18* represent the number of receipts in 2019 and 2018, respectively. *UniqueArticle19* and *UniqueArticle18* represent how many different types of articles were purchased (i.e., purchasing two apples and four tomatoes is accounted as two distinct articles and six products) for 2019 and 2018, respectively. The *PurchasedArticles19* and *PurchasedArticles18* represent the total number of products purchased in each year, regardless of whether they were fruits or vegetables. The other two variables, *UniqueArticles_Receipt* and *PurchasedArticles_Receipt*, represent the number of articles per receipt and the number of products per receipt, respectively, in each of the given periods of the year. Additionally, we computed the percent variation of the variety and quantity of fruit and vegetable articles per receipt for each variable (i.e., *VarUniqueArticles_Receipt = (UniqueArticles_Receipt19 − UniqueArticles_Receipt18)/UniqueArticles_Receipt18* and *VarPurchasedArticles_Receipt = (PurchasedArticles_Receipt19 − PurchasedArticles_Receipt18)/PurchasedArticles_Receipt18).* These two variables were the ones of interest, and they were analyzed to understand the impact of the *Nudge* intervention in each segment and month, and in the overall sample of the study.

## 4. Results

The data were analyzed using a one-sample *t-test* with a confidence level of 95% to understand which variables were significant and therefore could be used to draw conclusions. Using this simple analysis method provides important insights for future interventions and improvements. The results are divided by segment and followed by an analysis of the overall results (see Figures 1 and 2).

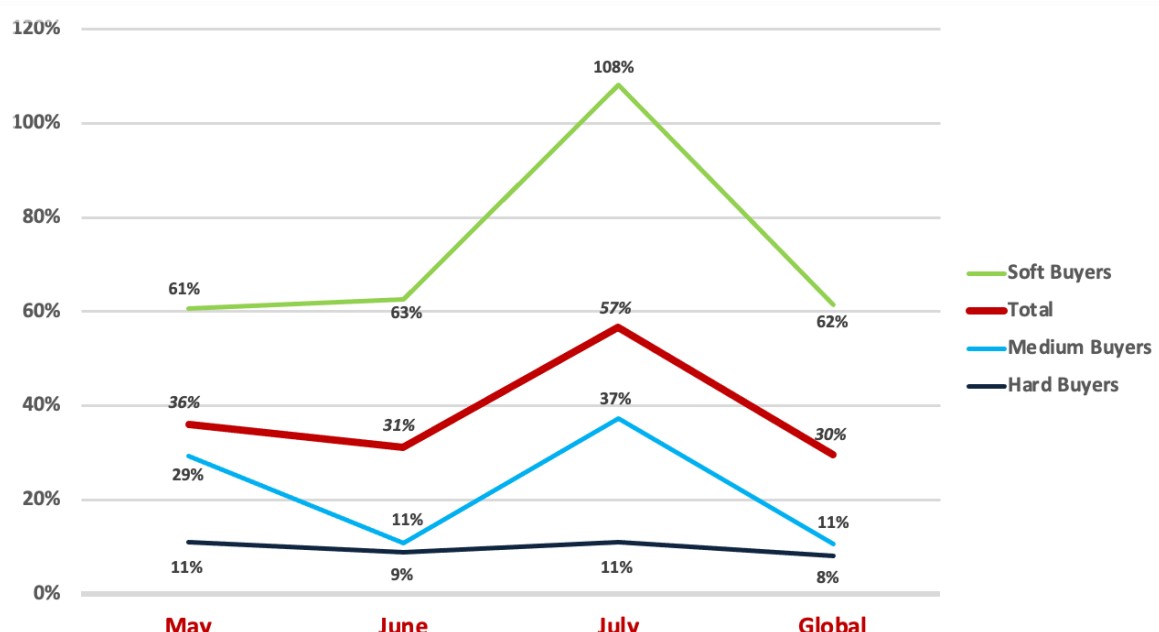

**Figure 1.** Variety percent variation per segment for all months.

According to the data obtained, using a one-sample *t-test* procedure, we could see that the impact of the intervention in terms of improving the variety of articles purchased was especially high among the *Soft Buyers,* with a Global increase of 62%, $t(1156) = 14.902$, $p = 0.000$. The *Soft Buyers* were followed by the *Medium Buyers*, by a significant amount, because the Global increase was only of 11%, $t(829) = 4.091$, $p = 0.000$, which was also very similar to the *Hard Buyers*, which had an increase of 8%, $t(968) = 4.572$, $p = 0.000$. Overall, the increase in the variety of articles purchased during the period of the intervention was 30%, $t(2995) = 15.455$, $p = 0.000$. In terms of months, the impact was higher during the month of July for all the segments, with the exception of the *Hard Buyers*.

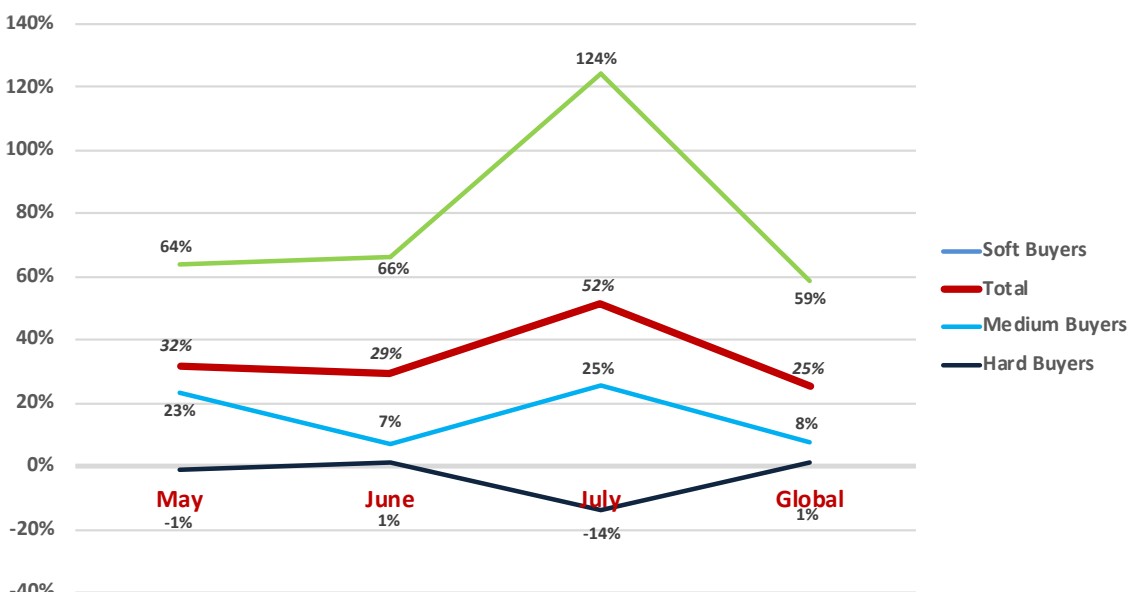

**Figure 2.** Quantity percent variation per segment for all months.

In terms of the number of articles of fruits and vegetables bought, we observed that the impact of the intervention was, again, particularly high among the *Soft Buyers*, with a Global increase of 59%, $t(1156) = 15.310$, $p = 0.000$, followed, by a significant amount, by the *Medium Buyers*, with a Global increase of 8%, $t(1156) = 3.146$, $p = 0.002$. In terms of months, the impact was also high during the month of July. It is notable that, among the *Hard Buyers*, the impact is next to zero and even negative during the month of July. Apparently, the fact that these buyers already regularly bought a number of articles greater than or equal to the number mentioned in the message (i.e., 11) neutralized the impact of the *Nudge* and even created a "boomerang effect" during the month of July.

In general, we obtained overall average increases in the variety of fruits and vegetables purchased of 30%, and overall average increases in the quantity of fruits and vegetables purchased of 25%, $t(2955) = 14.176$, $p = 0.000$. We also found significant variations in terms of the impact of the nudge intervention across segments of customers (the *Soft Buyers* were the ones most affected) and across months (an increase in the impact of the intervention as time goes by, with emphasis on the last month—July). Thus, after a careful analysis of the results, evidence was found to support both Hypothesis 1 and Hypothesis 2; not only was the intervention able to increase the quantity and variety of articles of fruits and vegetables purchased, it was also more impactful on the *Soft Buyer* purchasers. Lastly, it is important to observe that most of the variations detected were highly significant from a statistical viewpoint (i.e., $p < 0.001$).

## 5. Discussion

We can state that the nudge intervention applied by Auchan positively affected customers at a general level, although not all were influenced in the same way. The aim of this research was to identify and understand what aspects could be improved for future nudge retail interventions, especially in the health context, based in the results of an in-field intervention. Retailers have started to look at nudge interventions as valid options that provide short-term results that can deliver win-win situations at a marginal cost. In our case, Auchan was able to increase sales of healthy products and, at the same time, increase the consumption of healthy products by their customers for two of three segments, while negatively influencing the segment that already purchased large amounts of fruits and vegetables. It is crucial for retailers to understand that the outcomes will never be equal for everyone, and that delivering the right message for the right segment will increase the positive impact of each nudge intervention.

### 5.1. Theoretical Implications

Results for the three segments over the period of May and the three months confirm the first hypothesis that segments are affected differently by the same nudge. The first two segments revealed an increase in the consumption for fruits and vegetables, with a stronger influence on the first segment compared to the second. As seen in previous experiments with social norms, the message that customers were exposed to represented a social norm for the consumption of fruits and vegetables for a specific supermarket, providing a stimulus to purchase an average number of 11 fruits and vegetables. Customers in the first segment were most positively influenced due to having the largest shortfall of all three groups, highlighting the power of social norms to change consumer behaviors [31]. For the second segment, customers already consumed a level of these products that was closer to the messaged average, leading to a smaller increase than in the previous group. It is possible to conclude in this case that a social norm had stronger positive effects when the gap of consumption was larger to the messaged average. As expected, the group that previously purchased high amounts of fruits and vegetables was negatively influenced. Most customers were already purchasing higher levels than the average messaged by the social norm, and adjusted their consumption behavior to a level closer to the social norm average. In this case, the messaged social norm led the segment to an unwanted effect, demonstrating that the same social norm had both constructive and destructive effects [38]. Apart from the number of products, the nudge intervention also conveyed an implicit social norm that boosted the variety of products purchased. Because the intervention intention was to increase the volume purchased, the "*and*" word (referring to both fruits "and" vegetables) in the signage also allegedly stimulated customers to buy differentiated products.

Although the intervention was performed over a short period of time, different possibilities and outcomes might have resulted had the intervention been applied during a longer period and in all supermarkets. Because nudges focus on improving individuals' and society's long-term interests [12], a longer intervention could result in an increase of purchasing overtime, possibly creating new healthy eating habits. Additionally, the "11 fruits and vegetables" could be changed to a smaller number, affecting the segment that is already above the messaged average even more.

This analysis provides a useful insight for retailers to understand that nudge interventions will always have positive and negative effects in different segments, and that retailers should adapt each intervention to the type of customer rather than targeting the entire population. With the adaptation to the different segments, it should be possible to maximize the positive effects and minimize the negative ones.

### 5.2. Practical Implications

Most nudge interventions seek to change unwanted behaviors and consequently boost the profits of the store in the long run, while changing customers' shopping behaviors without restricting their options. In the retail context, several aspects should be assessed by retailers who seek to increase profits on the sales of healthy foods through the use of nudges; customers tend to be more influenced to make healthy food choices during the act of buying food rather than by exposure to information prior to the purchase. Interventions such as those taken at Auchan remind the customer to make healthy food choices from the beginning of the purchasing journey until the checkout. It is also possible to enhance nudges' effect by changing several store features such as the visual display of the food, lighting, or placement. Another important factor is the size of the store, which can influence the number of products purchased in each visit [56]. Therefore, if the Auchan intervention were executed in a smaller store, it is very likely that the percentage of Soft- and Medium Buyers would rise, and the intervention would have a greater positive impact.

Growing evidence suggests that changing food habits toward more healthy diets requires interventions in other social dimensions besides increasing levels of nutritional literacy. In the context of food-territory nudge interventions, tools/instruments that can "stir"

people's decisions can include association/articulation with other interventions, namely communication and pricing. A recent study conducted in a virtual supermarket with 346 Dutch participants reports interesting outcomes resulting from different combinations of discounts on healthy products, price increases in unhealthy products, and nudging [57].

Price promotions usually produce short-term effects, building a perception in consumers' minds of a good opportunity, especially if it relates to the basic basket. In Portugal, where the retail market is subjected to heavy price promotions, the effects of a price intervention may show enormous variation.

According to in-house data from the retailer, the most common pricing interventions (percentage discounts promoted on TV and in leaflets) produce different results regarding the quantity of a food product sold, depending on whether they pertain to processed foods, fresh products in general, or fruits and vegetables, in the range of 262% to 90%. The analytics are based on a direct comparison of the units sold during the duration of the promotion in leaflets (usually every two weeks) and the performance of the products at baseline (i.e., in the previous 28 days). These results do not reflect other types of promotions in nature or duration. Although the dimension of the sales increases, with the nudge intervention the effect is qualitatively different: changing habits, even if of short duration, vs. an opportunistic decision from the clients, without the need of reducing prices. In this way Auchan was able to achieve higher sales volume without cutting into margins in the fruit and vegetable category. Moreover, analyzing the intervention effects based only on the number of fruits and vegetables does not reveal the true extent of the intervention, especially if customers are in fact replacing unhealthy foods with healthier foods, as wanted, or simply continuing to purchase unwanted foods while adding healthier foods to the cart. From a business perspective, it is expected that profits on unhealthy foods will fall, revealing that the intervention is effective, but will be offset by increases in profits on healthy foods, though this remains to be confirmed.

Given the significance of the total results over the three months, it is possible to test and verify the second hypothesis in order to understand what the impact would be if the nudge intervention were to be applied to all supermarkets of the retail chain, that is, influencing the annual 18 million tickets containing at least one product of the category fruits and vegetables. With the 25% increase in the number of fruits and vegetables obtained, it is expected that the increase in the total number of articles purchased per receipt would be, on average, 0.75 (*PurchasedArticles_Receipt18* $\bar{x}$ = 3 × 0.25). When applied to the whole supermarket chain, that is, for the 18 million receipts, Auchan could potentially be selling an additional 13.5 million (0.75 × 18M) items of fruits and vegetables.

In light of these results, it is recommended that food retailers should adjust similar interventions to the size of each supermarket and understand what kind of customer they are targeting. In this specific case, an *injunction norm* messaging that high levels of fruit and vegetable consumption brings health benefits could reduce the negative results in the larger group, delivering even better results in the overall analysis.

*5.3. Limitations and Future Research Recommendations*

To the best of our knowledge, this is the first study in a "real world" setting where such a high number of clients and their performance have been evaluated. Several limitations can, however, be summarized.

As the *signage* was placed on the supermarket cart, it is very difficult to account for all the participants of this study due to the obvious limitation of gathering the data of thousands of people. Although the number of participants in this intervention was much larger than those accounted for, it was decided that using the data from customers' loyalty cards would be the best viable option to extract and further analyze the purchases, as this allowed comparing the purchases of the exact same clients in the same time frame of two different years. One more limitation arises from the fact that customers who did not use supermarket carts were therefore not exposed to the nudge but were nevertheless included in this study if they used their customer-loyalty cards.

Despite using an in-field intervention for the analysis, there are other limitations of the research that should be taken into consideration when interpreting the results. The Auchan supermarket for this study was located inside a shopping center in the center of Lisbon, in a district where the purchasing power of customers is above average, meaning that factors such as social class, education level, and income levels could affect the results. Another important factor is the season of the year in which the study was undertaken, as people shift their healthy shopping behavior depending on the time of the year. At the end of the year, especially from October to December, people make less healthy shopping choices than after January 1st, when they start to prepare for the summer [58]. It is possible that, because the study is based on data from a month before summer, the analysis could have been influenced by this factor. Another limitation is also present with the use of isolated *t-tests*; in our analysis, using multiple *t-tests* can increase the statistical error, leading to wrong results and interpretations. Even though this aspect can influence outcomes and recommendations, it is expected that the analysis is close to the real results.

Regarding the replication of our findings in other physical settings and other food categories, some of the outcomes of this intervention should be regarded with care, mainly due to: (i) fruits and vegetables is a category that is strongly associated with the rationalities of healthy eating, and it remains to be seen if the levels of adherence to the norm achieved in this case can be extrapolated to other food categories; (ii) The hyper/supermarket territory is influenced by layers of dimensions that are difficult to control. This means that direct comparisons about purchase performance between two supposedly similar time windows should be seen with caution because of the non-controllable, external variables such as economic environment, promotion campaigns by direct competitors, and product availability (very pertinent for fruits and vegetables, which are highly dependent on weather conditions).

Apart from the store and analysis limitations, it is difficult to predict similar outcomes in other supermarkets due to the demographic and economic limitations of each region. The same can be said for international interventions because cultures differ from country to country. A level of eleven fruits and vegetables may already be considered normal in other countries, leading to a larger negative effect or the other way around. Additionally, infinite reproduction of the intervention would likely lead to a normalization of the consumption until a point that it would have almost zero effect.

Despite a large amount of literature regarding the concept of nudge and nudge interventions, there are few studies that analyze real world situations, as herein. It is much more important to understand the changes in consumer behaviors in stores rather than sporadic experimental testing. Using data from store interventions could bring powerful insights depending on the industry and consumer types, helping governments and managers to make decisions about how to improve customer experience and behaviors.

Moreover, the existence of long-term analysis in this topic is minimal. With a tool like nudge, which has such a powerful influence on behaviors, long-term analysis of data can provide valuable insights, allowing retailers to adjust interventions to obtain the desired results.

## 6. Conclusions

Nudges seem to be a powerful tool to change and promote sustainable behaviors in society, with minimal associated costs. This paper tested evidence and demonstrated that, in a retail context, it is possible to deploy strategies that can effectively change purchasing behaviors. Healthy lifestyles often start at the supermarket, and that is the exact place to influence people's consuming behavior. Although nudge interventions have limitations, their reach and effectiveness taken alone or as part of a complementary strategy seem to be greater than most marketing techniques. This opens a new window for retailers to change customers' habits and boost general well-being in society.

**Author Contributions:** Conceptualization, D.G. and P.M.; methodology, D.G.; validation, D.G., L.F.M., and P.M.; formal analysis, D.G. and P.C.; investigation, D.G., P.C., L.F.M., and P.M.; resources, D.G., L.F.M., and P.M.; data curation, D.G. and P.C.; writing—original draft preparation, D.G., P.C., L.F.M., and P.M.; writing—review and editing, D.G., L.F.M., and P.M.; supervision, D.G. and L.F.M.; funding acquisition, D.G., L.F.M., and P.M. All authors have read and agreed to the published version of the manuscript.

**Funding:** This work was funded by Fundação para a Ciência e a Tecnologia (UID/ECO/00124/2013, UID/ECO/00124/2019 and Social Sciences DataLab, LISBOA-01–0145-FEDER-022209), POR Lisboa (LISBOA-01–0145-FEDER-007722, LISBOA-01–0145-FEDER-022209) and POR Norte (LISBOA-01–0145-FEDER-022209).

**Institutional Review Board Statement:** Ethical review and approval were waived for this study, as there this is a field study with no explicit manipulation rather than the nudge stimuli at the supermarket, which is similar to any advertising.

**Informed Consent Statement:** Patient consent was waived, as participants were unaware of this field study. Participants were primed only with the nudge stimuli at the supermarket, which is similar to any advertising. Moreover, they were not asked to answer any survey nor they were assigned to any experimental condition.

**Conflicts of Interest:** The authors declare no conflict of interest. The funders had no role in the design of the study; in the collection, analyses, or interpretation of data; in the writing of the manuscript, or in the decision to publish the results.

## Appendix A

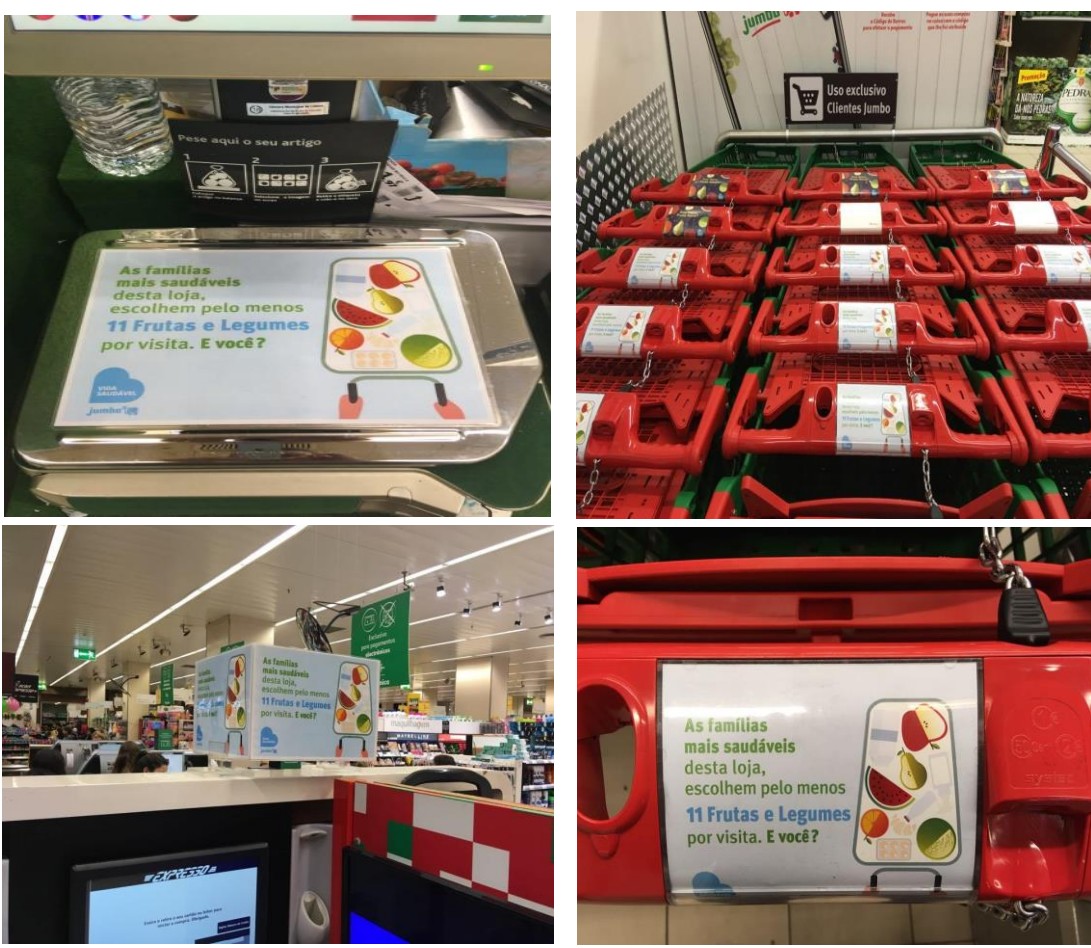

**Figure A1.** Examples of signage in the shopping carts and store.

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
