# Peer review of "Nudging Consumers toward Healthier Food Choices: A Field Study on the Effect of Social Norms"

_sustainability, doi:10.3390/su13041660_

Round 1

Reviewer 1 Report

The authors address an important topic: increasing people’s health through better nutrition choices. Overall, the article is well written. However, before the paper could be published some major revisions are required. Especially, the contribution of the study (in comparison to previous research) does not come across clearly in the introduction and theory part. Further shortcomings are the missing hypotheses, the use of adequate statistical methods and a more detailed description of results.

Title:

I would advise the authors to adapt the title to clarify that the goal is healthy food choices and that the intervention is social norms, e.g., as follows:

  • “Nudging with social norms: Increasing consumers’ healthy food choices”
  • “Nudging consumers towards healthier choices: A field study on the effect of social norms”

Introduction:

  • The authors highlight the practical relevance of the study (i.e. obesity), but do not outline the research gap. The authors simply stated that it is “important to understand how nudging can affect behaviors”, which is too unspecific and does not acknowledge the large body of research that has already examined the effect of nudges on various behaviors. The key research question should be included in the introduction and that further research is needed to answer it.

Theory:

  • 3: The difference between descriptive and injunctive norms is not fully clear in this section. Please add an example or highlight the difference in brackets (e.g. descriptive: “what most others do, what is done”; injunctive: “societal standards  for  how  people  should  act”/ “what ought to be done”).
  • As in the introduction, a reflection of prior research is missing here. The authors do not sufficiently describe what previous studies have been carried out. Several studies already analyzed the effectiveness of descriptive norm information, see examples listed below. It is absolutely key to summarize findings of previous studies on this topic and explain why further research is needed.

Studies on the effectiveness of descriptive norm information (food context):

  • Burger, J. M., Bell, H., Harvey, K., Johnson, J., Stewart, C., Dorian, K., & Swedroe, M. (2010). Nutritious or delicious? The effect of descriptive norm information on food choice. Journal of Social and Clinical Psychology, 29(2), 228-242.
  • Mollen, S., Rimal, R. N., Ruiter, R. A., & Kok, G. (2013). Healthy and unhealthy social norms and food selection. Findings from a field-experiment. Appetite, 65, 83-89.
  • Robinson, E., Thomas, J., Aveyard, P., & Higgs, S. (2014). What everyone else is eating: a systematic review and meta-analysis of the effect of informational eating norms on eating behavior. Journal of the Academy of Nutrition and Dietetics, 114(3), 414-429.

  • I would advise the authors to integrate the hypotheses (i.e. no separate chapter 2.4). Currently, the chapter is called “hypotheses development” but actually no hypotheses are mentioned in this chapter. The hypotheses should be clearly stated, for instance referring to the effect of different buying groups: “H1: Descriptive norm information increase healthy choices of unhealthy eaters.”; “H2: Descriptive norm information does not increase healthy choices of healthy eaters.” Or directly propose a hypothesis of the moderating effect of healthy eating habits: “H1: Healthy eating habits moderate the effect of descriptive norm information on food choices, such as healthy eaters are less affected.”
    The authors should also explain why they expect such an effect: you could argue this based on the findings by Schultz et al. (Study #33 cited on p. 3, l. 135-139).

  • Chapter 2.3 does not fit well into the red thread of the paper. In chapter 2.2, the authors are more specific (discussing the effect of descriptive/injunctive norms). Yet, chapter 2.3 is quite broad, discussing all kinds of nudge interventions. Changing the focus to a broader topic is not helpful and distracts from the actual research goal of the study.

Method:

  • Please explain why you chose “eleven fruits and vegetables” in the stimulus (why 11?)
  • Control period: maybe something changed between the years (2018/2019), did the authors also check the results in comparison to a month before the intervention took place (April 2019)?

Results:

  • Comments on Table 1: the definition of “hard buyers” group is inconsistent (in the text, p. 4: “more than nine articles”, in Table 1: “> 10”); What do the numbers in the table mean? Why do you show percentages in the first column, but not the other? What does “global” mean?
  • Comment on Figures 1 and 2: Quality should be improved (look blurred); Please explain what the numbers mean (i.e., what does “32% variety percent variation” mean?)
  • Several one-sample t-tests are used – however, due to the number of groups and months, many tests have to be conducted, which might lead to a multiple test problem (alpha inflation). The authors mention this in the limitation but did not check if this has affected the results or use another analysis method (see suggestion in the next comment).
  • Statistical tests should be conducted that explicitly test the hypotheses. For instance, the authors could conduct pooled regressions. By clustering errors by cases, the model takes into account that errors are correlated (same individuals are studied across time). Then, the authors could create dummy variables for the intervention (with/without), the time (month) and the buying habits (soft/medium/hard buyers) and test the effects.

Minor comments:

  • p. 1, l. 34: Delete „but“ at the end of the first sentence

Author Response

Reviewer #1

The authors address an important topic: increasing people’s health through better nutrition choices. Overall, the article is well written. However, before the paper could be published some major revisions are required. Especially, the contribution of the study (in comparison to previous research) does not come across clearly in the introduction and theory part. Further shortcomings are the missing hypotheses, the use of adequate statistical methods and a more detailed description of results.

Response: Thank you for your comments and interest in our paper.

Title:

I would advise the authors to adapt the title to clarify that the goal is healthy food choices and that the intervention is social norms, e.g., as follows:

  • “Nudging with social norms: Increasing consumers’ healthy food choices”
  • “Nudging consumers towards healthier choices: A field study on the effect of social norms”

Response: We agree the title could be improved to clarify both the research goal and the intervention. According to your suggestions, the paper is now retitled ‘Nudging consumers towards healthier food choices: A field study on the effect of social norms’.

Introduction:

  • The authors highlight the practical relevance of the study (i.e. obesity), but do not outline the research gap. The authors simply stated that it is “important to understand how nudging can affect behaviors”, which is too unspecific and does not acknowledge the large body of research that has already examined the effect of nudges on various behaviors. The key research question should be included in the introduction and that further research is needed to answer it.

Response: Thank you for your comment. Our intervention is based on a local context (one supermarket). We believe the research gap is now more explicit.

Theory:

  • 3: The difference between descriptive and injunctive norms is not fully clear in this section. Please add an example or highlight the difference in brackets (e.g. descriptive: “what most others do, what is done”; injunctive: “societal standards  for  how  people  should  act”/ “what ought to be done”).

Response: We agree that is a very important theoretical distinction, and we tried to better clarified it in the literature review.

  • As in the introduction, a reflection of prior research is missing here. The authors do not sufficiently describe what previous studies have been carried out. Several studies already analyzed the effectiveness of descriptive norm information, see examples listed below. It is absolutely key to summarize findings of previous studies on this topic and explain why further research is needed.

Studies on the effectiveness of descriptive norm information (food context):

  • Burger, J. M., Bell, H., Harvey, K., Johnson, J., Stewart, C., Dorian, K., & Swedroe, M. (2010). Nutritious or delicious? The effect of descriptive norm information on food choice. Journal of Social and Clinical Psychology29(2), 228-242.
  • Mollen, S., Rimal, R. N., Ruiter, R. A., & Kok, G. (2013). Healthy and unhealthy social norms and food selection. Findings from a field-experiment. Appetite, 65, 83-89.
  • Robinson, E., Thomas, J., Aveyard, P., & Higgs, S. (2014). What everyone else is eating: a systematic review and meta-analysis of the effect of informational eating norms on eating behavior. Journal of the Academy of Nutrition and Dietetics, 114(3), 414-429.

Response: We acknowledge the importance of the suggested references and have included them in the literature review.

  • I would advise the authors to integrate the hypotheses (i.e. no separate chapter 2.4). Currently, the chapter is called “hypotheses development” but actually no hypotheses are mentioned in this chapter. The hypotheses should be clearly stated, for instance referring to the effect of different buying groups: “H1: Descriptive norm information increase healthy choices of unhealthy eaters.”; “H2: Descriptive norm information does not increase healthy choices of healthy eaters.” Or directly propose a hypothesis of the moderating effect of healthy eating habits: “H1: Healthy eating habits moderate the effect of descriptive norm information on food choices, such as healthy eaters are less affected.”
    The authors should also explain why they expect such an effect: you could argue this based on the findings by Schultz et al. (Study #33 cited on p. 3, l. 135-139).

Response: Thank you for this pertinent comment. We clarified the reasoning behind our prediction, but we opted for not including explicit hypotheses, as we prefer to state our claims in the main text.

  • Chapter 2.3 does not fit well into the red thread of the paper. In chapter 2.2, the authors are more specific (discussing the effect of descriptive/injunctive norms). Yet, chapter 2.3 is quite broad, discussing all kinds of nudge interventions. Changing the focus to a broader topic is not helpful and distracts from the actual research goal of the study.

Response: We sought to remove too generic nudge interventions and include only those pertinent to our research topic, namely food and health interventions.

Method:

  • Please explain why you chose “eleven fruits and vegetables” in the stimulus (why 11?)

Response: We choose eleven fruits and vegetables as this number of items/products refers to the threshold that includes the top 10% receipts (i.e., percentile 90) for the actual consumption of these products. We also now include this explanation in the text.

  • Control period: maybe something changed between the years (2018/2019), did the authors also check the results in comparison to a month before the intervention took place (April 2019)?

Response: This is a good suggestion. However, the retailer mentioned no changes regarding consumption changes in that specific period, so we did not include this comparison in our results. We thus believe that did not impact significantly our conclusions.

Results:

  • Comments on Table 1: the definition of “hard buyers” group is inconsistent (in the text, p. 4: “more than nine articles”, in Table 1: “> 10”); What do the numbers in the table mean? Why do you show percentages in the first column, but not the other? What does “global” mean?

Response: We have now corrected this inconsistency between the text and the table.

  • Comment on Figures 1 and 2: Quality should be improved (look blurred); Please explain what the numbers mean (i.e., what does “32% variety percent variation” mean?)

Response: We apologize for the poor quality of the figures and hope the new versions are clearer.

  • Several one-sample t-tests are used – however, due to the number of groups and months, many tests have to be conducted, which might lead to a multiple test problem (alpha inflation). The authors mention this in the limitation but did not check if this has affected the results or use another analysis method (see suggestion in the next comment).

Response: Although we acknowledge the limitation of the use of multiple t-tests, this has not affected the global results nor the overall perception on these findings.

  • Statistical tests should be conducted that explicitly test the hypotheses. For instance, the authors could conduct pooled regressions. By clustering errors by cases, the model takes into account that errors are correlated (same individuals are studied across time). Then, the authors could create dummy variables for the intervention (with/without), the time (month) and the buying habits (soft/medium/hard buyers) and test the effects.

Response: We believe the t-test analysis is sufficient to confirm our predictions. Although the suggestion for conducting a pooled regression is pertinent, unfortunately we are not able to proceed with it due to time restrictions. Moreover, we believe the interpretation would not be significantly different from the one we used.

Minor comments:

  • p. 1, l. 34: Delete „but“ at the end of the first sentence.

Response: Thank you for highlighting this typo. We meant “(…) but, on the other hand, (…)” and this is now corrected. Once again, thank you for your insightful comments.

Reviewer 2 Report

The article “Nudging in the Food Retail Context: Changing Fruits and Vegetables Purchasing Habits Using the Human Urge to Conform” reports on a field-experiment aimed at nudging consumers towards healthier purchases using social norm.

Overall I liked the paper and the way Authors discuss their study, both in terms of strengths and in terms of limitations.

I only have some minor suggestions:

  • Page 3 lines 131-132: I don’t understand this sentence, in particular the pronoun it in “it will exert a much stronger influence on the behavior”: does it relate to the social norm that is present or to the absent one?
  • The quality of figures is quite poor and could be improved (but maybe it’s only temporary for the peer reviewing version).
  • Page 8 lines 337-338: I do not understand how the study proved that “the “and” stimulated to buy differentiated products”. First, it is not clear which “and” in the sentence “The healthiest families of this store purchase at least eleven fruits and vegetables per visit. And you?” the Authors are discussing: is it the one between fruits and vegetables or the final “and you”? And, secondly, how can they prove that the effect was stimulated by this specific word and not by the overall sentence? This point should be clarified.
  • It would be interesting to compare the effect of the intervention with some benchmark: eg, the 25% average increase in the quantity of fruits and vegetables purchased could be compared (with the same population) with the variation of number of food items (not only fruits and vegetables) purchased in the same period, or to the number of items  purchased in general. I expect that these benchmark variations would be minimal, but it would be interesting, both if they were low (it would mean that the increase in fruits and vegs decreased other purchases, i.e. an unintended outcome of the intervention) and if they were higher (in this case it would mean that the effect of the intervention could be a bit overestimated because consumers also increased in general their purchases  for other reasons, beside the nudge).

Author Response

Reviewer #2

The article “Nudging in the Food Retail Context: Changing Fruits and Vegetables Purchasing Habits Using the Human Urge to Conform” reports on a field-experiment aimed at nudging consumers towards healthier purchases using social norm.

Overall I liked the paper and the way Authors discuss their study, both in terms of strengths and in terms of limitations.

Response: Thank you for your suggestions and interest in our paper.

I only have some minor suggestions:

  • Page 3 lines 131-132: I don’t understand this sentence, in particular the pronoun it in “it will exert a much stronger influence on the behavior”: does it relate to the social norm that is present or to the absent one?

Response: We agree the above sentence is not sufficiently clear. We rephrased it to “In the case of individuals that already rank below the descriptive norm, the use of injunctive norms (i.e., referring to what is accepted or disapproved by society) can prevent them from increasing the undesirable behavior”.

  • The quality of figures is quite poor and could be improved (but maybe it’s only temporary for the peer reviewing version).

Response: We apologize for the poor quality of the figures and hope the new versions are clearer.

  • Page 8 lines 337-338: I do not understand how the study proved that “the “and” stimulated to buy differentiated products”. First, it is not clear which “and” in the sentence “The healthiest families of this store purchase at least eleven fruits and vegetables per visit. And you?” the Authors are discussing: is it the one between fruits and vegetables or the final “and you”? And, secondly, how can they prove that the effect was stimulated by this specific word and not by the overall sentence? This point should be clarified.

Response: We refer to the “fruits ‘and’ vegetables” quote, thus highlighting two healthy product categories. Additionally, although we acknowledge the effect of the overall sentence, we assume this specific “and” (on fruits and vegetables) will highlight the importance of buying more diverse healthy products. We hope this is now clearer in the text.

  • It would be interesting to compare the effect of the intervention with some benchmark: eg, the 25% average increase in the quantity of fruits and vegetables purchased could be compared (with the same population) with the variation of number of food items (not only fruits and vegetables) purchased in the same period, or to the number of items purchased in general. I expect that these benchmark variations would be minimal, but it would be interesting, both if they were low (it would mean that the increase in fruits and vegs decreased other purchases, i.e. an unintended outcome of the intervention) and if they were higher (in this case it would mean that the effect of the intervention could be a bit overestimated because consumers also increased in general their purchases for other reasons, beside the nudge).

Response: Thank you for your pertinent comment. However, we feel this is out of scope for this research. Eventually future research could address that specific issue. Once again, thank you for your pertinent comments.

Reviewer 3 Report

The paper titled Nudging in the Food Retail Context: Changing Fruits and Vegetables Purchasing Habits Using the Human Urge to Conform describes the results from a nudging intervention in a Portuguese supermarket, aimed at increase the purchasing of fresh fruit and vegetables.

The results are based on a solid data collection, in terms of quantity and methodology used for the cellection, and the potential limitations of the results are well presented in the discussion section.

However, two main aspects of the research should be enriched.

The first aspect is related to the introduction section. Since the study aims to investigate the effect of a nudging intervetion on the selling of fresh fruit and vegetables, a more detaile description of the actual condition of the Portuguese market for fresh fruit and vegetables, should be presented. Also, a comparison of the actual quantities of fruit and vegetables consumed in Portugal with the recommendation from International Organizatios (e.g. WHO) should be useful to have a more clear overview of the Portuguese situation. 

The second aspect concerns the Result sections. While the results of the t-test -are quite well presented, the inferential analysis cited in the abstract is missing, and it seems to be reported only in the conclusions (lines 387-394). A more detailed description of the inferential statistical analysis of the data and of the model adoted to obtain the results presented in the aforementioned part of the discussion section could strengthen the discussion section and the practical implications of the study. Also, the authors should be careful in generalizing too much the results of the inferential analisys to the general Portuguese context, because of the limitations of the study well presented in the conlusions section.

Good luck with your submission!

Author Response

Dear Sir/Madam,

             Thank you very much for reading our paper and for the helpful feedback. We implemented the changes you suggested in the first aspect of your review, through the use of 2 new references. 

             Regarding the second aspect, we had more difficulties in adding new information to the article, since a detailed description of the inferential method used is already described in the conclusions (Lines 404 to 411), were we felt it made more sense, in terms of helping to enlighten the practical implications of the study.

Thank you very much for your help, Diogo Gonçalves 

Round 2

Reviewer 1 Report

Thank you for your response and for your revisions. Overall, the authors have addressed some of the questions in the first round of review. I agree with the changes of the title and the research contribution added in the introduction.

However, I think the authors have not adequately addressed the points that I wanted the authors to focus on, namely the missing hypotheses and improving the statistical analysis.

  • The authors simply ignored my advise to state specific hypothesis, yet the reasoning for this is not clear to me. Given the plethora of previous research in this area, it can be expected by the authors to formulate their predictions in testable hypotheses. I even made suggestions how the hypotheses might be formulated, hence addressing this comment should have been not too hard.
  • The results part is still quite short and the authors still do not really explain in the text what the numbers in the figures mean. Further, if they use t-tests, the results should be presented in an appropriate way, i.e., reporting dfs, the t statistic and p values (see e.g., the APA reporting standards)
  • The authors claim that they could not conduct a pooled regression due to time restrictions. However, this argument is not convincing to me: if the deadline for the revision was too ambitious, they might have asked the editor for an extension. Conducting a pooled regression is a quite common, standard statistical analysis, hence it could have been expected by the authors to run a regression analysis. 

Additional comments:

-quality of figure 2 is improved, yet figure 1 is still blurred

Overall, I still believe the study as such could offer some valuable insights, yet the way the authors present it currently do not meet key scientific standards: formulating hypotheses, testing them with appropriate statistical methods, and presenting the results in a detailed, transparent way.

Unfortunately, the authors did not carefully address my comments from the first review round, which pointed out these shortcomings. Before the paper could be published, I would advise to at least add hypotheses and test, which of them can be accepted or rejected (and fully report the according test statistics).

Author Response

Dear Sir/Madam,

          We implemented all the changes you suggested, with the exception of the Regression Analysis. We are happy with the final result: we feel the paper improved. The reason for not implementing the regression analysis is due to the fact that its not the most suitable analysis for data obtained via quasi-experimental method analysis, and also because it would make the results analysis to large, without adding significantly understanding to the analysis already done with the T-test analysis.

Thank you very much for your feedback, Diogo Gonçalves 

Reviewer 3 Report

The integrations made to the manuscript improved it and filled the gaps that were present in the previous version of the manuscript. Thanks and good luck with your submission!

Author Response

Thank you! 

Diogo Gonçalves

Round 3

Reviewer 1 Report

The integrations made to the manuscript improved it, yet before publication I have minor comments to the added parts:

First, I would suggest to rephrase the hypotheses. I would recommend to explicitly name the type of intervention (descriptive norm) and the comparison group. Also, terminology should be consistent (e.g. “soft buyers” as in p.6/Table 1).

“H1: A descriptive norm intervention in the store will increase the purchase of fruits and vegetables (compared to a period without intervention).”

“H2: Fruit and vegetable purchases will increase more among soft/medium buyers (compared to heavy buyers with high fruit/vegetable purchase).”

Second, please clarify some aspects about the segmentation (section 3.3.) in the paper:

  • Based on which time period did you cluster the customers? Based on purchases in 2018 or 2019?
  • Please clarify in Table 1 what the numbers mean. Are these the number of customers or the number of articles bought?

Third, concerning the results part:

  • Instead of adding a general sentence at the end of the results part (l. 329ff.) that the hypotheses are supported, please include this information directly after the respective statistical test that supports the hypotheses. Currently, it is not fully clear which test supports which hypothesis.

General comment: please refer to all Tables and Figures are least once in the text.